# Moving towards Integrated and Personalized Care in Parkinson’s Disease: A Framework Proposal for Training Parkinson Nurses

**DOI:** 10.3390/jpm11070623

**Published:** 2021-06-30

**Authors:** Marlena van Munster, Johanne Stümpel, Franziska Thieken, David J. Pedrosa, Angelo Antonini, Diane Côté, Margherita Fabbri, Joaquim J. Ferreira, Evžen Růžička, David Grimes, Tiago A. Mestre

**Affiliations:** 1Department of Neurology, University Hospital Marburg, 35033 Marburg, Germany; thieken@staff.uni-marburg.de (F.T.); david.pedrosa@staff.uni-marburg.de (D.J.P.); 2Cologne Center for Ethics, Rights, Economics, and Social Sciences of Health (CERES), University of Cologne, 50931 Cologne, Germany; Johanne.Stuempel@uk-koeln.de; 3Research Unit Ethics, University Hospital Cologne, 50931 Cologne, Germany; 4Parkinson and Movement Disorders Unit, University of Padua, 35122 Padua, Italy; angelo.antonini@unipd.it; 5The Ottawa Hospital Research Institute, Ottawa, ON K1Y 4E9, Canada; dianedenis0719@gmail.com; 6Department of Neurosciences, Clinical Investigation Center CIC 1436, Parkinson Toulouse Expert Center, NS-Park/FCRIN Network and NeuroToul COEN Center, TOULOUSE University Hospital, INSERM, University of Toulouse 3, 31062 Toulouse, France; margheritafabbrimd@gmail.com; 7Laboratory of Clinical Pharmacology and Therapeutics, Faculdade de Medicina, Universidade de Lisboa, 1649-028 Lisboa, Portugal; jferreira@medicina.ulisboa.pt; 8Instituto de Medicina Molecular João Lobo Antunes, Faculdade de Medicina, Universidade de Lisboa, 1649-028 Lisboa, Portugal; 9CNS—Campus Neurológico Sénior Torres Vedras, 2560-280 Torres Vedras, Portugal; 10Department of Neurology and Center of Clinical Neuroscience, First Faculty of Medicine, Charles University, General University Hospital in Prague, CZ-121 08 Prague, Czech Republic; evzen.ruzicka@lf1.cuni.cz; 11Parkinson Disease and Movement Disorders Centre, Division of Neurology, Department of Medicine, The Ottawa Hospital Research Institute, University of Ottawa Brain and Mind Research Institute, Ottawa, ON K1Y 4E9, Canada; dagrimes@toh.on.ca (D.G.); tmestre@toh.ca (T.A.M.)

**Keywords:** Parkinson’s disease, nursing training, integrated care, Parkinson nurse, personalized care, multidisciplinary care

## Abstract

Delivering healthcare to people living with Parkinson’s disease (PD) may be challenging in face of differentiated care needs during a PD journey and a growing complexity. In this regard, integrative care models may foster flexible solutions on patients’ care needs whereas Parkinson Nurses (PN) may be pivotal facilitators. However, at present hardly any training opportunities tailored to the care priorities of PD-patients are to be found for nurses. Following a conceptual approach, this article aims at setting a framework for training PN by reviewing existing literature on care priorities for PD. As a result, six prerequisites were formulated concerning a framework for training PN. The proposed training framework consist of three modules covering topics of PD: (i) comprehensive care, (ii) self-management support and (iii) health coaching. A fourth module on telemedicine may be added if applicable. The framework streamlines important theoretical concepts of professional PD management and may enable the development of novel, personalized care approaches.

## 1. Introduction

Parkinson’s disease (PD) is a progressive non-curable neurodegenerative disorder with an age of onset usually over 60 and presenting with complex motor and non-motor features such as cognitive impairment, mood and sleep disorders, autonomic dysfunction, and pain. In Europe, 1.2 million people are living with PD [1] with an increasing incidence in the elderly, so that the number of affected patients worldwide is expected to double by 2030 [2]. PD ranges among the top ten most resource intensive brain disorders in Europe [1] so that the need for PD services is expected to build up and consequently the burden on healthcare systems. This reality warrants the development and implementation of a care delivery model that conforms with society resources to guarantee its sustainability, while promoting better public policies, and reducing the overwhelming societal impact of PD [3]. The complexity of PD implies specific requirements for the design and delivery of care. Nevertheless, to date personalized care delivery models are rare [4]. While it has been shown that integrated and multidisciplinary care delivery models following a personalized care approach have positive implications for persons living with PD (PwPs), care partner and care providers, their implementation is difficult due to several reasons [3,4,5]. A key aspect for personalizing care services, is the availability of specialized staff [3]. Among these healthcare professionals, Parkinson Nurses (PN) can accomplish important tasks in the care process, such as providing mental health support, monitoring symptom progression and promoting patient navigation through the local healthcare system [3,6,7]. There has not been a consensus on defining the PN, but following Parkinsons UK, a PN can “…provide expert care because they only work with people with the condition.” They describe the major role of a PN in providing care as whilst “…helping people to manage their medication” [8], the provided care by a PN will result in less side effects. Generally, PN help patients to manage their illness through making, for example by giving information and support to people with Parkinson’s.

However, there are various definitions and descriptions not only of the role in the care team but there are also multiple approaches on the training of PNs as highlighted in Table 1.

Even though specialized training for PN on the delivery of personalized care services has been recommended [17], no framework has been proposed yet and existing curricular do not explicitly in cooperate it. By reviewing the specific requirements for the design and delivery of care in PD, we aim to propose a training framework for PN to facilitate the personalization and integration of care delivery.

## 2. Materials and Methods

We adopted a conceptual research approach to synthesize different perspectives on the theme of PD care and role of nurses. [18]. We entertained various conceptual streams from health care design, care delivery and medical prerequisites of PD. We considered the following questions to be essential to PN training: What are the care priorities for people living with PD (PwPs)? What type of PD-specific skills should a PN be equipped with in order to meet these needs?

The conceptualization of a training framework, may thus be seen a synthesis from various theoretical concepts which address care priorities for PD. We followed the structure of a line of reasoning to model this novel concept [19]. In this approach different hypothesis are formulated which then are integrated into a proposed model [19]. Following Lynham’s Growth Cycle of Applied Theory-Building, the conceptualization of the training framework was informed by research, theory and practice [20]. A scoping literature review was conducted in order to identify relevant literature on care for PD. We chose The methodological approach of a scoping review, as it has been recommended to be particularly useful for categorizing the existing scientific literature in a defined research area in terms of its type, characteristics, and scope [21].The literature review was conducted in March 2020, (with an update in April 2021) by searching MEDLINE and Web of Knowledge (Figure 1), using the terms Parkinson’s disease, concept and care. The search was not restricted in terms of the publication year. Studies were included if they described or theorized care models or concepts, relevant to PD and were either published in English or German. If a paper referred to another theory which was not focusing on PD but still relevant for the research aim (developing a PN training framework), the paper and theory reported also included. Opinion papers, literature reviews not proposing a new care model or studies testing short term interventions (i.e., physical therapy) were excluded. Publications focusing on palliative care were excluded, because this was seen as a different topic, where PwPs and care partners develop unique needs and concepts become relevant, which distinguish from other PD care literature. The search strategies, as well as a detailed list of in- and exclusion criteria can be accessed in the Appendix A.

As a first step, publications retrieved from the literature review were grouped into 3 categories: intervention, practical care concept or theoretical concepts. Next, practical care concepts, guidelines and interventions were reviewed to identify care priorities for PD. Consecutively, a code was invented, whenever a new guiding care principle was mentioned following the approach of an undirected content analysis [22]. Thirdly, the identified care priorities informed the formulation of two hypothesis regarding a PN training framework. Fourth, the content of theoretical concepts was analyzed according to the previously identified care priorities. Finally, the content was used to construct a line of reasoning and to propose a framework for training PN. The literature research and coding, following the guidelines of PRISMA-ScR [23], was independently performed by two researchers (M.vM; J.S.). Discrepancies in coding and grouping were solved via discussion with a third researcher (F.T.). The final framework was commented by a range of PD experts for the iCARE-PD consortium (http://icare-pd.ca/, 1 June 2021), including PN, neurologists and scientists.

## 3. Results

Fifty-six publications were included for final synthesis. (Figure 1).

The analysis included nine interventions implementing and evaluating a care model for PD, two guidelines for organizing PD care, 29 publications describing an implemented care model and 18 conceptual papers. Few publications described the same practical care model [24,25,26,27] whereas two reported the same care concept [28,29]. Consequently, 35 publications informed the definition of care priorities for PD care and 18 models informed the conceptualization of a PN training framework. Based on the implemented care models and recommendations, nine priorities for the organization and delivery of PD care were identified. The priorities and the frequency with which they were mentioned are summarized in Table 2.

The conceptual models covered the same care priorities as the practical care concepts. In addition to these priorities, the priority *personalized care* was observable in the conceptual models. The priorities and the frequency with which they were mentioned are summarized in Table 3. A content summary of the included models can be found in the Appendix A.

Based on the identified care priorities for PD patients and their conceptualization in various care models, we present two hypotheses on the training requirements for PN, followed by relevant question(s) related to each hypothesis and their implication to the development of a PN training framework.

**Hypothesis** **1** **(H1).**
*Parkinson Nurses should be trained to deliver comprehensive care for people living with parkinson’s Disease and their care partner.*


Given the heterogeneous and progressive nature of PD, treatments require a high degree of personalization, as this enables the adjustment of the multiple existing management options to the clinical presentation, the individual symptoms and their progression, and the care needs of PwPs [81]. Based on the analyzed concepts of PD care, two models described personalized care management as important aspect [70,75] whereas two other models included the provision of tailored information [69,80].

What is personalized care? Personalizing care means adapting the care process to the patients’ needs and preferences [78] (813) (p.813). Van Halteren et al. described five essential aspects of personalized care: providing information, proactively monitoring early detection signs and symptoms and the care process, coordinating care and navigating the patient in the healthcare system [78].

Implications for a PN curriculum: In reference to the conceptualization of a training framework, a PN ought to be competent to identify care needs and preferences for each individual. Additionally, they must be able to decide their implications for the care plan.

Personalizing care approaches means, that patients’ perspective plays a central role in decision-making processes and leads to another frequently mentioned care priority: patient-centered care. Two models incorporated patient-centeredness as a pivotal aspect for care delivery [66,70] and three models highlighted the patients perspective as central component [65,68,71].

What is patient-centered care? Implementing a patient-centered perspective means ‘*[…] ensuring that patient values guide all clinical decisions*.’ ([69], p. 360). Good communication is needed in order to identify these values [64,67]. PN must be able to meet patients and care partner with respect and empathy [64,67,69,72]. Providing emotional support and creating a trustful relationship has been mentioned as important element for implementing patient-centered care across all three identified concepts [67,69].

Implications for a PN curriculum: In reference to the conceptualization of a training framework, a PN ought to be trained in communicating with PwPs and care partner to enhance patient-centeredness.

What is integrated care? Integrated care is a form of multidisciplinary care. A multidisciplinary care approach can be described as an approach ‘[…] with contributions by experts from multiple complementary disciplines.’ [49] (p.167). Bringing together these professions is what Goodwin described as professional integration [74]. Other concepts referred to this by highlighting the importance of incorporating physicians’ perspectives in the care process; coordinating care across professions and implementing a clinical information system [64,68,77].

While there is a wide range of definitions, integrated care can be described as a care approach that aims ‘[…] bringing together key aspects in the design and delivery of care systems that are fragmented’ [74] (p 1) (p.1). Three conceptual models described components that an integrated care approach should consider [66,74,76]. The Rainbow defines four primary domains of integration: clinical, professional, organizational, and systems integration, whereby functional and normative enablers play a role [76]. The Development Model of Integrated Care (DMIC) presents a nine-cluster model for organizational development in four phases with an emphasis on actual co-operation and commitment [66]. The DMIC also focuses on conditions for achieving effective collaboration, such as patient engagement, clarity of roles and responsibilities within the care delivery team [66]. Goodwin’s work [74] distinguishes not only in the form in which integrated care should be designed (horizontal, vertical, sectoral, people-centered and whole-system), but also by how it is classified (by type, level, process, breadth and degree/intensity) [74].

An important aspect that was identifiable across the three integrated care concepts is care organization [66,74,76]. For PD, the inclusion of multiple healthcare professionals and the coordination of their care actions is of utmost importance [3]. Delivering integrated care has been described as central aspect for meeting PwPs complex care needs, reducing the burden of care partner and improving health care professional satisfaction [4].

Two of the conceptual models in integrated care services as important aspect for care delivery [73,75] and three models referred indirectly to the integration of care by mentioning a continuous collaboration of care providers, the organization of care and the selection of combined helping methods as important aspects of care organization [68,77].

Implications for a PN curriculum: PN fulfill important roles as clinical care integrators, navigators, support person and supervisor [4,75,82]. PNs, as part of the professional care team, should be able to design and implement a flexible routine network of service provider to support PwPs and their care partner in inpatient and outpatient settings.

What is home-based and community-centered care? Home-based care ‘*[…] refers to clinical practices that provide physician- or nurse practitioner led, longitudinal interdisciplinary care […]*’ at home ([79], p. 1). According to the Quality of Care Framework for Home-Based Medical Care [79], the essential elements are: assessment, care-coordination, patient and care partner education, provider competency, safety, provider competency and shared decision-making [79]. Additionally, factors such as patient and care partner experience, financial aspects and quality of life should be considered [79]. According to the model, patient-centered care can be promoted through the use of quality indicators that assess patients’ access to care services, as well as their satisfaction with the expertise of care providers [79]. From the reviewed conceptual models, two referred indirectly to the organization of home-based care by mentioning the support of autonomy as important aspect for organizing patient-centered care [55,56]. Three models highlighted the need to assess available community resources [58,68], one model referred to the importance of assessing the personal lifestyle [59], one model defined quality criteria for the implementation of home-based telemedicine [19,20] and five models mentioned the navigation of the patient towards these resources and the reduction of barriers as important aspect for the organization of care and the selection of combined support methods as important aspects of care organization [59,60,62,68,69].

Delivering community-centered care means bringing ‘*[…] care directly to the patients in the local community setting […].*’ ([30], p. 1). Consequently, knowledge about the community and available resources is required. Based on the literature review, no model exclusively focusing on community-centered care was identified, however several concepts included available community resources as important quality aspects of care [67,77] as further detailed above.

Delivering care at home and within the community is important for PwPs and their care partners in order to enable access to care [3,73,83]. Additionally, home-based care for PwPs is becoming increasingly important from a demographic (e.g., aging, immobile population) and social (e.g., patients having a pronounced desire to continue living in their own homes) point of view [68].

Implications for a PN curriculum: Based on the concepts of home-based and community-centered care, we propose that the quality of care provided by PN may be influenced by level of coordination skills of different stakeholders in the healthcare system and knowledge about local healthcare resources. Thus, PN should be trained to map available community resources and navigate PwPs towards them.

**Hypothesis** **2** **(H2).**
*Parkinson Nurses should be trained to deliver self-management support to persons with Parkinson’s Disease and their care partner.*


Self-management support (SMS) and patient-education are critical elements of effective PD management [4,84], and key component of integrated care. SMS is a top priority for PwPs when asked about their care requirements [85]. SMS and patient-education help to reduce disease progression, complications and costs [4,84,85].

What is patient and care partner education? Based on Graham’s concept, patient and care partner education are a form of knowledge translation [80]. The ability of lifelong learning is an important aspect of healthy aging and may be jeopardized by PD [70]. Implementing learning processes and empowering patients and care partner through education characterize integrated care concepts that were included in the analysis [66,69,75,77]. According to the Knowledge Translation Framework, patient education should be based on identified problems and adapted knowledge based on these problems. Patients and care partner should be motivated to use the delivered knowledge. Additionally, the identification of barriers and the use of knowledge should be evaluated continuously [80].

What is self-management support? Self-management support ‘*[…] aims to empower patients with the skills and confidence necessary to manage their clinical disease*.’ ([73], p. 25). Activities include patient education, monitoring changes in symptoms and abilities, goal setting, and problem-solving [73]. Based on Orem’s Self Deficit Theory, self-management support is needed, when the client’s self-care demand exceeds the available self-care agency [68]. From the literature review, four conceptual models were identified that included self-management support as important aspect of care [65,70,73,77]. The Glasgow model (or 5-A’s approach) describes five important actions that should be taken by the health-care professional when delivering SMS to the patient, namely: assessing, agreeing, advising, arranging and assisting. Another model, which is often referred to by SMS interventions for PwPs is the Chronic Care Model [77]. The model does not exclusively focus on SMS, but describes SMS as one of six dimensions, which should be addressed to improve care for patients with a chronic disease. According to the model, all dimensions affect each other, which is why all dimensions should be considered when aiming to improve care. The Chronic Disease Self-Management Model [77] is another model, which does not explicitly address PwPs but informed SMS approaches for PD [86]. Similar to the Glasgow model, it focuses on the relationship between the healthcare professional and the patient, however, a stronger focus is placed on the motivational aspect. According to the model, a good SMS-program pays attention to emotional and role management in addition to medical management and incorporates techniques to improve the patients’ confidence.

Implications for a PN curriculum: PN play an important role in delivering SMS to PwPs [4,87,88], as good SMS relies on support from educated health professionals [88]. PN have a have a close patient contact and thus, are ideal professionals for delivering SMS [88]. In order to advise and assist PwPs properly, an understanding of the disease and its complexity is required, making it an essential part of a PN training framework. Considering the Knowledge Translation Framework, we propose that a PN training should include aspects of motivational interviewing in order to facilitate knowledge use by PwPs and their care partners [89].

Finally, one of the identified theoretical concepts considered telemedicine [28,29]. Telemedical applications can improve PwPs access to care, enhance quality of life and reduce the burden of care partner [90]. However, their purpose can vary greatly [28], which is why we propose to add a fourth module to the PN training framework when applicable, specifically focusing on the available technology.

### Proposing a Framework for Training Parkinson Nurses to Deliver a Personalized Care Approach

In the previous section, we have formulated two hypotheses: (1) PN should be trained to deliver comprehensive care for PwPs and their care partner and (2) PD Nurses should be trained to deliver self-management support to PwPs and their care partner. Based on the review of conceptual models, we identified the following requirements to a framework for training PN:(1)PN ought to be competent to identify needs and preferences. Additionally, they must be able to decide their implications for the care plan.(2)PN require training in communicating with PwPs and care partner.(3)PNs, as part of the professional care team, should be able to design and implement a flexible routine network of service providers to support PwPs and their care partner in inpatient and outpatient settings.(4)The quality of care provided by PN may be influenced by specific training in the coordination of different stakeholders in the health care system and knowledge about local healthcare resources. Thus, PN should be trained to map available community resources and navigate PwPs towards them.(5)In order to advise and assist PwPs properly, an understanding of the disease and its complexity is indispensable, making it an essential part of a PN training. Considering the Knowledge Translation Framework, we propose that a PN training should include aspects of motivational interviewing in order to facilitate knowledge use [89].(6)Education on telemedicine should be incorporated whenever possible and applicable.(7)Based on these requirements, we propose that PN should be trained in three central aspects in order to deliver a personalized care approach: i. understanding PD, ii. health coaching and iii. delivering comprehensive care. These aspects form the framework of the PN training displayed in Table 4.

Understanding the disease is a fundamental prerequisite for delivering care and, consequently, a foundational knowledge and skills for PN training [6,17,82]. PN must be able to adapt care delivery to the care requirements of PwPs and care partner, which change across the course of the disease. After completing the first module, PN are equipped with skills, that are important for integrating, personalizing and centering care around the patient. Besides a sound medical knowledge, PN must be able to understand and conduct clinical assessments [91]. These assessments may help the PN to evaluate patient needs as a starting point for discussion about care plans. Additionally, aspects of patient education and self-management support come into play when the PN discusses tests results or care plans with the patients. Consequently, we propose training on clinical assessments. And obtaining clinical conversation skills as central goal for the second training module: health coaching. Optimal care of PD should promote general health and wellbeing and care priorities should be defined together with PwPs and care partner [3]. Also, PwPs and care partner require a reference person that can be embodied by the PN through the empathic assessment of their care needs and the nurse’s role as a care coordinator [3]. After the completion of this second training module, PN will have acquired the skills to assess personal care requirements of PwPs. The understanding of PD and health coaching skills merge, in line with the care priorities of home-based and community centered care, into a third and last module: delivering comprehensive tailored care. PwPs and care partner have to be navigated throughout the local healthcare system; multiple professions have to be incorporated in the care process and PwPs and care partner need motivation to use these resources. Consequently, we propose that PN should be trained to identify relevant local resources for PwPs and care partner, understand their living situation and motivate them to utilize available resources. Finally, a fourth module regarding available technologies can be added if applicable. This module will be discussed in greater detail in the following section.

## 4. Discussion

This paper proposes a framework for a novel PN training in the context of integrated care. There is a scientific consensus that PNs will take a significant and prominent role in integrated, patient-centered home-/and community-based care in the future [92]. The PN is widely considered to be an important primary point of contact for PwPs and care partner alike. PNs are also recognized to be very helpful in the role of a multidisciplinary care team coordinator [93]. When PNs are available to provide home-based care, it has been shown that patients’ quality of life improves [54,94]. The importance of professional education can be identified in both theoretical and intervention-based models [55,65].

When it comes to educating PN, a variety of training pathways exist in the various countries. Also, the recognition of nurses as important care coordinators differs. As it has been stated elsewhere, funding mechanisms and the structure of healthcare systems play an important task for defining a nurse’s role [95]. This is also reflected in different education programs. Therefore, it is necessary to address country specific requirements when implementing the framework. Also, the structure of healthcare systems affects the availability of resources, which is why a sound understanding of the overall context is essential for implementing the framework presented here. We emphasize that module 3 of the framework should be adjusted to the country-specific context. Further research may aim to further defining this module and adapting it to a country-specific context. For countries with extensive training opportunities and high resources, such as the United States [95], we suggest, that single modules of the proposed training framework could be implemented in the basic training of nurses as prerequisite for a later specialization in the field. This would enable nurse students to better understand PD and prepare them to be empowered nursing advocates for PwPs in inpatient and outpatient settings. For countries, where the profession of PN is less well developed, such as Germany [7,95] the framework may be fully implemented and also be utilized to build an agenda for future research on how the role of PN can be strengthened.

When implementing the framework into pratice, one might face challenges and barriers. In some countries, the PN has been an integral part of the multidisciplinary care team for a long time [96], while in others PN are not present in every care team [7,17]. Also, it is necessary to clarify funding issues for implementing the framework and hire staff, such as experienced PN, to deliver the framework. Additionally the lack in certification of such training could be another barrier [7]. However, the framework introduced here may represent an crucial step towards a universal consensus on certification.

Concerning the fourth module, the framework is deliberately kept open. Telemedicine represents an increasingly studied and apparently beneficial instrument for the provision of medical care to the chronically ill [29,97,98]. However, telemedicine must always be evaluated in the context of its application, i.e., the technical prerequisites for widespread use must also be accessible to the individual patients [99]. Therefore, telemedicine is not yet part of this framework, but we strongly encourage its future integration. Due to the emerging possibility of remote patient monitoring (i.e., smart glasses, smart beds or wearables [83,100]), we emphasize future research on up-to-date tech-based home-based care solutions and the future role a PN may hold in this scenario of increasingly tech-based medical and social care delivery. This demand would also meet the need of care approaches to not only being responsive to specific care situations, but to incorporate proactive elements, such as the utilization of telemedicine [92,93].

For the future, the model being proposed here should configure a practical care concept that addresses effectively the identified care priorities for PD. One important aspect is the validation of the role of the PN and its training across cultures and societal contexts. Further research may focus on evaluating the implementation of the framework into a practical care concept and the development of a toolkit, which allows a flexible and streamlined adaptation of the training curriculum into different settings. Also, the model may be extended by reviewing care priorities for palliatve care.

## 5. Limitations

This review holds potential limitations. The quality of evidence, which was included in the review was not assessed, since the purpose was to review existing concepts as widely as possible. Further, the curriculum has not been implemented or evaluated in practice, which is why no claims about its feasibility can be made. Rather, it should be understood as stimulating and inspiring source of information for developing future PN curricula. Finally, the framework does not include country-specific differences of PN, which may affect its applicability.

## 6. Conclusions

A training framework for PN introduced here marks a pivotal contribution to increase the quality of care delivery for PwPs and their care partner following a care priority adapted approach. This framework is intended as an invitation to other researchers and practitioners to aid supporting the role of PNs and to move towards a standardized training. A shift towards a proactive role of a PN amongst healthcare providers is necessary and should be encouraged by legislation.

## Figures and Tables

**Figure 1 jpm-11-00623-f001:**
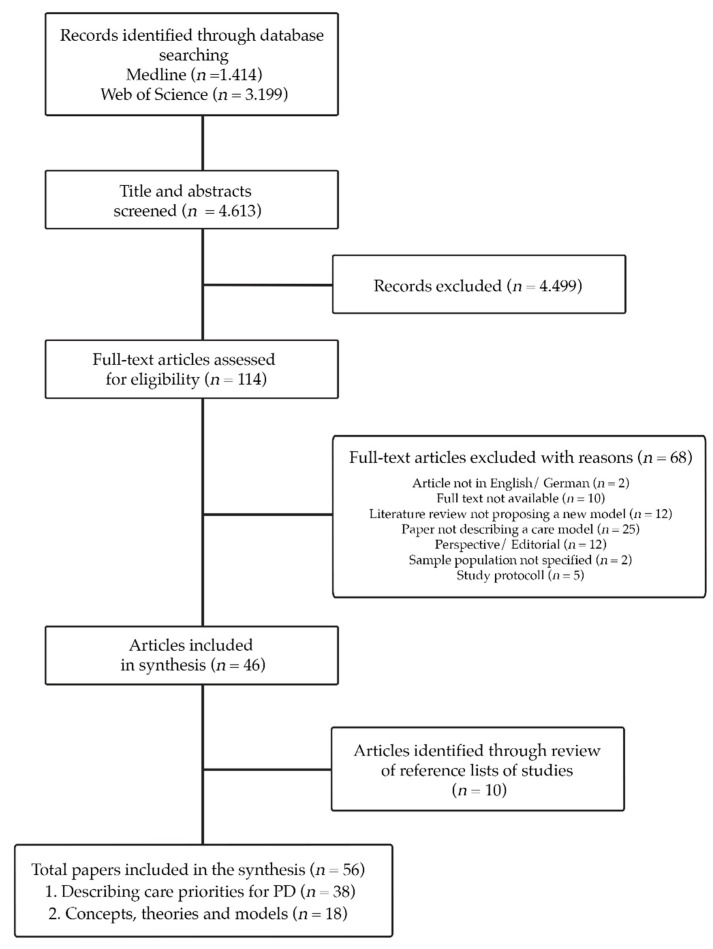
PRISMA flow diagram for the conducted scoping review.

**Table 1 jpm-11-00623-t001:** Existing Training Opportunities for Parkinson Nurses in different Countries.

Country	Role	Formal Education	Reference
United Kingdom	Being responsible for overall management within primary or secondary care teams Resource of Information and advice for PwPs Catalyst for improving public awareness	Provided via national universities Prerequisite: Being a registered Nurse (registered no longer than three years) Being registered at the Nursing and Midwifery Council (NMC) Proving high level of experience working with and managing PD or other neurodegenerative disease Topics covered during education: Strengthen experience through active involvement in clinical care Education about principles of primary and secondary care Responsibilities in care for PwPs Aspects of multidisciplinary care coordination	[8,9]
Germany	Providing information and advice to patients and care givers on medication, symptoms and treatment options	Provided via German Parkinson Society (DPG), German Parkinson Association, (dPV), Parkinson Competence Network (KNP), Association of Parkinson Nurses and Assistants (VPNA) Prerequisite: Completed 3-year regular nursing training + at least 2 years of working experience in the field Topics covered during education: Specialist knowledge on special treatment procedures (e.g., medication pump or Deep Brain Stimulation) Psychological counseling Activating therapies Handling of specific medications for PwPs	[7,10]
United States ^1^	Role of APN (generally) Care of patients Education Research Consultation Leadership They have: Expert knowledge Decision-making skills Clinical competencies APNs are trained to work autonomously in specific care areas.	Advanced Practice Nurse (APN) → post-graduate education in nursing Two Types of APN roles have been recognized in the United States. Nurse Practitioner (NP) Clinical Nurse Specialist (CNP) Provided via national universities & national council Prerequisite: Bachelor’s degree in nursing + passed national council licensure Examination For advanced nursing: master’s degree in Nursing + specialty education Topics covered during education: Advanced health assessment Diagnosis Disease management Health promotion Health prevention Evaluation Research	[11,12]
Canada ^1^	Role of APN (generally) Care of patients Education Research Consultation Leadership Role of NP Legal authority to provide diagnosis and/or interpret diagnosis tests Prescription of medication Perform interventions In Alberta, British Columbia and Ontario: Admission and discharge of patients from the hospital NPs work in primary and community care setting Responsible for health promotion, disease prevention, the diagnosis and management of acute illness and the management of the chronically ill Role of CNP: Multi-faceted Variable Deployment in clinical care, education and research Involvement in organizational leadership and professional development Providing evidence-based practice and efforts on program development	Two types of APN roles have been recognized in Canada. Nurse Practitioner (NP) Clinical Nurse Specialist (CNP) Provided via national universities & national council NP → Registered nurses; completed NP education program; Bachelor- or Master’s degree CNP → Master or Doctoral degree in nursing Topics covered during education for APNs: Leadership Accessibility of care Safety of delivering care Plan, coordinate, implement and evaluate programs to meet patients’ needs Promotion of community health	[13,14,15,16]

^1^ In the United States and Canada, a variety of different forms of nurse education exist, and the nomenclature also holds a wide range of designations in both countries. The role and education of APNs (Advanced Practice Nurses) will be discussed here as an example. An explicit training as an advanced practice nurse for Parkinson’s disease is not currently available in the United States and Canada.

**Table 2 jpm-11-00623-t002:** Care Priorities for Parkinson’s Disease in Practical Care Concepts.

Care Priority	Citation (Frequency)	Reference
Multidisciplinary care	24	[19,24,25,26,27,30,31,32,33,34,35,36,37,38,39,40,41,42,43,44,45,46,47,48,49,50,51]
Patient-centeredness	17	[24,25,26,27,32,36,39,40,41,42,43,44,45,47,48,52,53,54,55,56]
Integrated care	16	[3,24,25,26,27,31,32,36,38,40,42,44,45,46,48,51,52,56,57]
Home-based care	13	[36,37,40,41,42,44,51,53,56,57,58,59,60]
Self-management	11	[24,25,26,27,36,39,40,44,45,55,57,61,62]
Community-centered care	9	[24,25,26,27,30,41,45,52,53,59,60,63]
Patient-/care partner education	7	[36,39,40,42,44,51,55]
Telemedicine	7	[30,42,44,56,57,58,59]
Professional education	1	[55]

**Table 3 jpm-11-00623-t003:** Care Priorities for Parkinson’s Disease in Conceptual Models.

Care Priority	Citation (Frequency)	Reference
Patient-centeredness	9	[64,65,66,67,68,69,70,71,72]
Integrated care	8	[66,68,69,73,74,75,76,77]
Multidisciplinary care	6	[64,66,68,74,76,78]
Community-centered care	6	[67,68,69,71,77,78]
Home-based care	5	[28,29,65,68,73,79]
Personalized care	4	[70,75,78,80]
Self-management	4	[65,70,73,77]
Patient-/care partner education	2	[75,80]
Telemedicine	1	[28,29]
Professional education	1	[65]

**Table 4 jpm-11-00623-t004:** Conceptual framework for training Parkinson-Nurses to deliver a personalized care approach.

Module	Topic	Components	Goals
1	Understanding Parkinson’s disease	Understanding Parkinson’s disease—symptoms and care requirements Parkinson’s disease stages and care needs Aspects of Parkinson’s disease management	Acquire fundamental knowledge about PD and management principles of motor and non-motor symptoms
2	Being a health coach	Clinical assessments for people living with Parkinson’s disease and care partners and their implications for care requirements Acquire understanding of what’s important when managing Parkinson’s disease in a day-to-day practice and at home Aspects of clinical conversations (identifying care priorities etc.)	Acquire skills and knowledge to assess patient-outcomes and identify personal care requirements
3	Aspects of care delivery for people living with Parkinson’s disease and care partner	Acquiring an overview of local care resources and important contact points for people living with Parkinson’s disease and care partner Building a local care network Conversation training to motivate patients and care partners Delivering self-management support based on the 5-As Learning about the role of care partners as support person	Acquire knowledge about available local care resources and methods to motivate patients and care partners to use them
4	Telemedicine	Adapted to the specific technology Identify role of technology in care model: online monitoring, self-management support, enhance communication	Acquire knowledge about the technology

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
