# Peer review of "Moving towards Integrated and Personalized Care in Parkinson’s Disease: A Framework Proposal for Training Parkinson Nurses"

_jpm, 2021, doi:10.3390/jpm11070623_

Round 1
Reviewer 1 Report
Revision of the article:
Moving towards integrated and personalized care in Parkinson´s disease: proposing a framework for training Parkinson Nurses
This article shows a framework for training Parkinson Nurses based on a previous review of the literature on care priorities for people living with Parkinson´s disease.
The subject of the article is very interesting and of great importance to train nurses so that they can respond to the needs of people living with Parkinson´s disease, but it would be desirable to be more specific in some of the points of the conceptual framework described in table 3, according to all the aspects described in the two hypotheses raised.
In addition, another important aspect to review is the references section. You can check the recommendations in: https://www.mdpi.com/journal/jpm/instructions
Kind regards
Reviewer 2 Report
The article is interesting and could be used not only for training the nurse but also the student. Limits of the study are not present
Reviewer 3 Report
The authors present a thoughtful systematic review of the literature with the goal of developing a framework for training Parkinson Nurses. This review adds to the literature, and will be helpful for anyone that is considering training or developing a training plan for Parkinson Nurses. The only concern is that although it appears multiple countries in Europe are represented, there are likely very different models for the care of a Parkinson patient based on the training level of the nurse and the organization of the healthcare system. For example, in the United states, there are multiple levels of training for nurses with widely diverging roles from a nurses aid, nurse assistant, registered nurse, advanced practice nurse, physician assistant, just to name a few. Given the goal of this manuscript, I would advise the authors to include a section defining what a Parkinson Nurse is, and how that can change in different healthcare systems. It is fine for this to be limited in scope to healthcare systems in Europe, but it does reduce the impact and applicability of this work.
Author Response
Dear Sir or Madam,
Thank you for your comments on the paper. Please find the table below summarizing the changes made in the article based on all reviewers' comments.
We hope that this will meet your expectations and remain at your disposal for any further information.
Kindest regards,
Marlena van Munster
Summary table of responses to the reviewer comments
|
Editor/Evaluator’ comments |
Authors' responses |
|
Reviewer 1: “…it would be desirable to be more specific in some of the points of the conceptual framework described in table 3, according to all the aspects described in the two hypotheses raised.” |
Thank you for raising this important point. We have addressed this important concern and, by rewriting some of the passages in the manuscript (starting in Section 3.1), we have now been able to further clarify how the framework described in Table 3 helps to fulfill/confirm the hypotheses we stated. |
|
Reviewer 1: “…another important aspect to review is the references section.” |
The references of the manuscript were carefully checked in great detail. The publisher's specifications were thereby met and the manuscript now complies with the requirements. |
|
Reviewer 2: “…Limits of the study are not present.” |
We have revised the conclusion-section of the manuscript. The limitations of the study are now summarized in a separate section and are therefore now considerably made more apparent to the reader. |
|
Reviewer 2: “The article is interesting and could be used not only for training the nurse but also the student.” |
Thank you for this suggestion, we included this in our discussion section. |
|
Reviewer 3: “The only concern is that although it appears multiple countries in Europe are represented, there are likely very different models for the care of a Parkinson patient based on the training level of the nurse and the organization of the healthcare system. For example, in the United states, there are multiple levels of training for nurses with widely diverging roles from a nurses aid, nurse assistant, registered nurse, advanced practice nurse, physician assistant, just to name a few. Given the goal of this manuscript, I would advise the authors to include a section defining what a Parkinson Nurse is, and how that can change in different healthcare systems. It is fine for this to be limited in scope to healthcare systems in Europe, but it does reduce the impact and applicability of this work.” |
Thank you for pointing out this important aspect. We added a definition defining what a Parkinson Nurse is in the introduction, as well as an exemplary overview about training opportunities in different countries (EU and non-EU). We also discussed the importance of adapting the framework to the requirements of different health care system and mentioned this as potential limitation. |
Please see the attachment

Round 2
Reviewer 3 Report
authors have dressed recommendations appropriately.